# *Rhus vulgaris* Meikle fruit-mediated silver nanoparticles: Synthesis, characterization, and potential for sun protection, antioxidant and antibacterial applications

Woinshet Kassie Alemu[1], Limenew Abate Worku[2]*, Rakesh Kumar Bachheti[3], Archana Bachheti[4]

**1** Department of Industrial Chemistry, College of Natural and Applied Sciences, Addis Ababa Sciences and Technology University, Addis Ababa, Ethiopia, **2** Debre Tabor University, College of Natural and Computational Science, Department of Chemistry, Debre Tabor, Ethiopia, **3** Centre of Molecular Medicine and Diagnostics, Saveetha Dental College & Hospitals, Saveetha Institute of Medical and Technical Sciences, Saveetha University, Chennai, India, **4** Department of Environment Science, Graphic Era (Deemed to be University), Dehradun, Uttarakhand, India

\* limenewabate@gmail.com

## Abstract

This study investigates the synthesis of silver nanoparticles (AgNPs) using *Rhus vulgaris* fruit extract (RVFE) and evaluates their antioxidant, antibacterial, and UV protection properties. *R. vulgaris*, a member of the Anacardiaceae family, is known for its rich phytochemical profile, including phenols, glycosides, alkaloids, saponins, tannins, and terpenoids, which contribute to its medicinal properties. The AgNPs were synthesized by mixing RVFE with silver nitrate ($AgNO_3$) under optimized conditions: a temperature of 80 °C, a pH of 9, a reaction time of 40 minutes, and a 9:1 $AgNO_3$-to-RVFE ratio. Characterization of the synthesized AgNPs was performed using UV-Vis spectroscopy, Fourier Transform Infrared (FTIR) spectroscopy, Transmission Electron Microscopy (TEM), X-ray Diffraction (XRD), and zeta potential analysis. The results confirmed the successful synthesis, with AgNPs exhibiting an average size of approximately $14.64 \pm 0.4$ nm and a zeta potential of -26.0 mV, indicating good stability. The antioxidant activity was assessed using the DPPH radical scavenging assay, revealing a maximum inhibition of $53.7\% \pm 0.12\%$ with an $IC_{50}$ value of 81.2 µg/mL. Antibacterial tests demonstrated significant activity against various bacterial strains, with inhibition zones ranging from $23.88 \pm 1.10$ mm to $30.21 \pm 1.21$ mm, indicating a stronger effect against Gram-negative bacteria. Additionally, the synthesized AgNPs exhibited a high Sun Protection Factor (SPF) of 234.5, suggesting their potential as effective UV blockers. This study highlights the promising applications of AgNPs-RVFE in biotechnology, medicine, and healthcare, emphasizing their eco-friendly synthesis and significant biological activities.

**Data availability statement:** All relevant data are within the paper.

**Funding:** The author(s) received no specific funding for this work.

**Competing interests:** The authors declare that there is no conflict of interest regarding publication of this paper.

## Introduction

The majority of important medicinal herbs belong to the family Anacardiaceae of plants. The secondary metabolites found in these plants can potentially treat several illnesses [1]. Rhus is a well-known taxon in the family, and several taxa have significant antibacterial, antifungal, antiviral, antioxidant, antimalarial, and cytotoxic properties [2]. *R. vulgaris is* one such wild edible fruit-producing plant under the family Anacardiaceae. It is mainly found in temperate, subtropical, and tropical regions of the world, especially in Africa, especially in all of Tanzania, Uganda, Kenya, Ethiopia, Mozambique, Malawi, Zambia, and Zimbabwe, and in North America [3]. Studies on *R. vulgaris'* phytochemistry have revealed a high concentration of bioactive compounds, such as phenol, glycosides, alkaloids, saponins, tannins, and Terpenoids [2,4]. The stem bark of this plant is used to treat malaria in Kenya [5]. This plant is one of the most widely used chewing sticks in Uganda and is used to alleviate toothaches [6]. *R. vulgaris* leaves have long been used in Tanzania to heal dental issues and as a refreshing beverage in herbal teas [7]. Fruits of *R. vulgaris* are used to cure gastrointestinal issues, syphilis, coughing, toothaches, and infections in the East African regions around Lake Victoria [8].

When provided as a crude extract, the bioavailability of the active bioactive chemicals is significantly decreased; however, when the crude extract is provided in a modified form, such as nanomaterial, this can be improved [9]. Nanostructures have been produced by biology, medicine, and microelectronics researchers, and nanoscience is gaining popularity these days [10]. Identifying, producing, and developing nanoparticles for application in pharmaceutical therapy are the main areas of focus for current research [11]. Both chemical and biological processes can produce nanoparticles. Plants, microbes, and enzymes are thought to be the basic components of biological nanoparticles [12].

Additionally, the therapeutic qualities of medicinal plants are increasing the value of the nanoparticles they produce because many medicinal plants have substances with antibacterial and antioxidant properties, such as phyto-compounds, which have additive potential with these nanoparticles [13]. Furthermore, compared to other forms of nanoparticles, the use of biological nanoparticles is thought to be favorable because they do not require the use of hazardous chemicals during their manufacture [14]. Furthermore, it was discovered that the synthesis procedure affected the unique characteristics of metal nanoparticles [15]. Depending on their intended uses, a variety of metal nanoparticles have been designed and manufactured. Based on the biological approach, metallic nanoparticles containing Au, Zn, Ag, and Cu numerous other elements have been synthesized [16] For instance Lava et al. [17] synthesized AgNPs using *Justica wynaadensis* leaf extract. The leaf extract solution reduced and stabilized the AgNO3 into AgNPS with size in the range from 30 nm to 50 nm. In another research work Basavegowda et al. [18] synthesized bimetallic ZnO–CuO hetero-nanocomposite, ZnO, and CuO nanostructure using leaf extract of *Aegle marmelos.* In addition to this Mishra et al. [19] a green and facile approach for synthesizing Fe, Pd and Fe–Pd bimetallic nanoparticles using an aqueous bark extract of *Ulmus davidiana.* Due to their unparalleled physical, chemical, electrical, and magnetic properties [10] and their toxic potential against pathogens like bacteria,

viruses, and fungi, AgNPs are regarded as a valuable option for nanomedicine applications like drug delivery [20], cancer treatment [21], and wound healing [22]. The biosynthesis of AgNPs involves the utilization of many extracts from plants that are commonly used in traditional medicine or that contain bioactive chemicals. Due to their reducing agent properties, the secondary metabolites included in various plant extracts are in charge of causing the nanoparticles' particle sizes to decrease [23]. The biological characteristics of nanoparticles can be improved by synthesizing them from plant extracts. As a result, biological techniques for producing nanoparticles from diverse plant extracts have become increasingly significant in various biological and therapeutic applications [24]. As antioxidants, antimicrobials, and sun blockers, AgNPs have found widespread application recently [25]. Because *R. vulgaris* is used as a medicinal plant [26] and as a food additive [26] and because there has not been any prior research on the plant's antimicrobial, sun screening, or antioxidant activity, *R. vulgaris* fruit extracts were used in this study to synthesize AgNPs-RVFE. When applied to the breast cancer cell line (Cell line A549), the biological activities of AgNPs demonstrated notable antibacterial and anticancer effects. The anti-inflammatory and anti-diabetic properties of *J. wynaadensis* AgNPs have also demonstrated their suitability as a substitute in both industrial and medical settings [17]. The main goal of this study was to create AgNPs-RVFE using fruit extract of *R. vulgaris* and to assess the antioxidants, antimicrobials and sun screaming activity of synthesized AgNPs-RVFE. The produced nanoparticles were characterized using various methods, including UV–vis spectroscopy, TEM, FTIR, and DLS.

## Materials and methods

### Sample collection and preparation

*R. vulgaris* Fruits were collated from the Amhara region, East Gojam zone, Mertulemariam Wereda, Ethiopia, located about 364 km from the capital city of Addis Ababa, Ethiopia. The collected plant materials were taxonomically identified by Dr. Melkamu Wondaferah, Department of Botany, and Addis Ababa University, Addis Ababa Ethiopia. The voucher specimens of the collected plant materials were deposited at the National Herbarium with voucher number 3074. The plant samples were collected when ready for consumption. After gathering the healthy Fruits, they were adequately cleaned using tap water to remove any remaining soil. After washing, the fruits were allowed to air dry for 15–20 days at room temperature in a shaded area, following the method described by Murthy et al. [27]. To perform the extraction, 450 ml of distilled water was combined with 50 g of powdered vegetable fruits in a 500 ml Erlenmeyer flask. The flask was covered with aluminum foil to protect the solution from light exposure. The mixture was shaken for 30 minutes using a mechanical shaker, and then heated on a hot plate with a magnetic stirrer at 80 °C for 1 hour before being allowed to cool to room temperature. The resulting solution was filtered through Whatman No. 1 filter paper to obtain a clear extract. The filtrate was then stored at 4 °C labeled as RVFE for use in subsequent tests.

### Silver nanoparticle synthesis

The synthesis of AgNPs-RVFE was optimized based on the method from Kumar et al. [28] with minor modifications. Different ratios of (9:1, 7:3, 5:5, and 6:4) aqueous fruit extracts of *R. vulgaris* to 0.04 M aqueous silver nitrate solution, respectively, were heated at different temperatures (70, 80, and 90 °C) for (30, 40 and 50 minutes) with constant stirring using magnetic stirrer using different pH conditions (4.5, 6, 7.5 and 9). The formation of the AgNPs-RVFE (reduction process of silver ions ($Ag_+$) to silver ($Ag^0$) nanoparticles) is preliminarily detected by the color change of the solution from pale yellow to dark brown or brownish-yellow depending on the reaction condition. Centrifugation was used for 30 minutes at 5,000 rpm to separate the produced AgNPs-RVFE. The finished synthesized AgNPs-RVFE are freeze-dried and kept in a dark, temperature-controlled environment at 4 °C until needed [28].

### Characterization techniques

Using UV/Vis Spectroscopy, the absorbance of synthesized AgNPs-RVFE was measured. A UV-Vis spectrophotometer can be used to assess the optical properties of biosynthesized AgNPs-RVFE samples at room temperature to track the

completion of Ag$_+$ bio-reduction in aqueous solution [29]. To track the reduction of pure Ag$_+$ ions, a small aliquot of the sample is diluted in distilled water, and the UV-Vis spectra of the reaction medium are recorded after a two-hour period. Furthermore, X-ray diffraction (XRD) methods are applied to characterize the plant-mediated AgNPs-RVFE produced during the synthesis. After centrifuging the material, pellets were created and ground into powder. The powder was placed within the X-ray diffraction (XRD) cubes, and the result was collected and examined in the XRD apparatus [30]. In addition to XRD, Transmission Electron Microscopy (TEM) is utilized to ascertain the size and shape of the nanoparticles using TEM (JEOL 2100F) operating at an accelerating voltage of 120 kV. A carbon-covered copper grid was coated with AgNPs-RVFE prior to TEM picture capture. Finally, AgNPs-RVFE are analyzed using Fourier Transform Infrared (FTIR) spectroscopy, which uses the KBr disc approach to create an IR beam of IR spectrometer scanning spectrum in the range of 550–4000 cm$_{-}^1$ at a resolution of 16 cm$_{-}^1$ [31].

### DPPH radical scavenging activity of AgNPs-RVFE

With slight modifications, the method described by Keshari et al. [32] was employed to evaluate the ability of AgNPs-RVFE to scavenge free radicals using the stable radical DPPH assay. Three milliliters of freshly prepared DPPH (0.1 mM in methanol) solution were mixed with 1 milliliter of optimized AgNPs (AgRvNPs$_4$) solution at concentrations of 20, 40, 60, 80, and 100 mg/ml, and the mixture was gently vortexed. The combination was then allowed to sit for 30 minutes at room temperature in the dark. A UV-Vis spectrophotometer was employed to measure the absorbance at 517 nm. Methanol served as the blank solution, while DPPH was used as the control to exclude the samples. Ascorbic acid (AA) is being used as a positive control. Equation 1 was performed to obtain the percentage of inhibition for the free radical scavenging activity. The antiradical activity is measured by calculating the percentage inhibition of DPPH radical following the addition of test samples, and this is expressed as a drop in absorbance.

$$Free\ radical\ scavenging\ activities\ (\%) = \frac{A_{DPPH} - A_{solution}}{A_{DPPH}} \times 100 \ldots$$

(1)

A$_{DPPH}$ and A$_{solution}$ denote DPPH absorbance at 517 nm before and after adding AgNPs-RVFE.

### Sun protection factor (SPF) determination

Three widely used and commercially available human hand and body lotions, including Yellow color Vaseline lotion (YVL), Aloe Vera Vaseline Lotion (AVL), and Rounsun Honey Lotion (RHL) purchased from a local market in Addis Ababa, Ethiopia. The Sun SPF determination was performed based on the methods used by Sayre et al. [33] with some modifications. 0.2 g of each commercial formulation lotion was added separately in a 100-mL volumetric flask, diluted to 15 mL with green solvent (water). Then, from these flasks, 2 mL of the lotions were taken individually, and AgNPs-RVFE synthesized using fruit extract of *R. vulgaris* (100 µg/mL) were added. The plant extracts, and the commercially available cosmetics (lotions) without AgNPs-RVFE were used as a control. Cotton was used to filter the final solution. Following a 15-minute incubation period, the solution was measured for absorbance in the wavelength range of 290–320 nm at 5 nm intervals. The Mansur equation (Equation 2) was then used. Triple-checked experiments and the resulting values were utilized to compute SPF.

The Mansur equation is used to determine the SPF of different formulations. Sayre et al (1989) [33] developed the equation as follows:

$$SPF = CF \times \sum_{290}^{320} EE\ (\lambda) \times I(\lambda) \times Abs(\lambda)$$

(2)

Where: EE (λ) = the erythemal effect spectrum,

I: solar intensity spectrum,

CF: correlation factor (10)

Abs: absorbance of sunscreen product

Value of EE ($\lambda$) × I is constant and is determined by Sayre et al. [33] (Table 1).

## Antimicrobial activity

Antibiotic susceptibility testing was conducted using disk diffusion method. The bactericidal activity of four different samples such $AgRvNPs_4 25$, $AgRvNPs_4 50$, $AgRvNPs_4 75$, and $AgRvNPs_4 100$ representing for 25, 50, 75 and 100 gm/ml respectively was examined. A sterile swab was used to inoculate a sterile agar surface with various Gram-positive bacteria (*S. aureus* and *K. pneumonia*) and Gram-negative bacteria (*E. coli* and *P. aeruginosa*). Using the disk diffusion method, Whatman No. 1 sterile disks, measuring 6 mm, were dipped in AgNPs-RVFE. The antibacterial properties of AgNPs-RVFE and the plant's crude extracts were compared. These disks will be sterilely air-dried before being put onto the MHA (Mueller Hinton Agar plates) seeded top layer and cultured for twenty-four hours at 37 °C [34].

## Statistical analysis

The values were expressed using the means ± standard error of the means. Using SPSS/20 software, a one-way analysis of variance was employed to do a statistical analysis of the data. A p-value of less than 0.5 was considered statistically significant.

## Result and discussion

### Silver nanoparticles biosynthesis

The AgNPs-RVFE were successfully synthesized using *R. vulgaris* fruit extract as both a reducing and capping agent, with silver nitrate serving as the precursor. This process was optimized through carefully tailored parameters, including temperature, concentration, pH, and reaction time, resulting in the effective formation of AgNPs. When the fruit extract was added, the colorless silver nitrate solution was seen to turn dark brown; in contrast, no color change was seen when the plant extract was added (Fig 1). The reaction mixture's color shift was one of the main pieces of evidence for creating AgNPs-RVFE [35]. Afterward, the produced AgNPs-RVFE underwent several characterization techniques.

In green synthesis, phytochemicals and secondary metabolites found in plant extracts are considered reducing agents. The phytochemicals found in plants facilitated the creation of metal nanoparticles, even if the complete picture of this process is still poorly understood. The reduced metal ions are connected by oxygen created during photochemical degradation. Phytochemicals stabilize them by preventing their aggregation [36]. To illustrate, Fig 1 demonstrates how the hydroxyl groups in the bioactive chemicals interact with the $Ag_+$ ions in the plant extract to generate AgNPs-RVFE, which also serve to stabilize the newly formed nanoparticles [37]. For instance, when flavonoids interact with metallic ions, the carbonyl functional groups in the extract release reactive hydrogen, which converts the flavonoid's enol into keto form and finally generates $Ag^0$ (Fig 1) [38].

Table 1. Normalized product function used in the calculation of SPF.

| Wavelength (nm) | EE x I (Normalized) |
|---|---|
| 290.00 | 0.0149 |
| 295.00 | 0.0800 |
| 300.00 | 0.2869 |
| 305.00 | 0.3280 |
| 310.00 | 0.1840 |
| 315.00 | 0.0827 |
| 320.00 | 0.0179 |

## Effects of operational parameters for the synthesis of AgNPs-RVFE

**Effect of pH on green synthesis of AgNPs-RVFE.** Using basic solution (1M NaOH) and acidic condition (1M HCl), the pH of the green generated AgNPs-RVFE optimization was adjusted to a final pH of 4.5, 6, 7.5, and pH 9. A double-beam UV-Vis spectrophotometer was used to analyze the absorbance of the AgNPs-RVFE. Table 2 shows that the absorption peaks were $0.185 \pm 0.36$, $0.537 \pm 0.26$, $0.618 \pm 0.75$, and $0.851 \pm 0.58$ for acidic circumstances (pH = 4.5 and pH = 6), nearly neutral conditions (pH = 7.5), and basic conditions (pH = 9), respectively, while maintaining other effects such reaction time, concentration, and temperature. At a pH of 9.0, the maximum absorption was noted. This demonstrated that while the synthesis of AgNPs-RVFE utilizing fruit plant extract is appropriate at a pH of 9.0, the least amount of absorption was seen in an acidic environment at a pH of 4.5 (Fig 2). This indicates that AgNPs-RVFE perform better in basic conditions (high absorbance) than in neutral or acidic conditions. The work of Gou et al. [39] confirmed these findings, demonstrating that the maximum absorbance increased as the pH ranged from 6 to 12. Their results indicated that an alkaline pH enhanced the production of AgNPs. Increased availability of bioreductants can lead to better generation of AgNPs-RVFE under alkaline conditions; this can be explained by a change in the dissociation constants of the functional groups of biomolecules involved in reducing $Ag_+$ ions. The study by Melkamu and Bitew [40] showed that basic conditions are favorable for AgNPs synthesis. It was found that basic pH of 8 and pH of 9 environments resulted in relatively larger AgNPs sizes with more significant wavelength shifts compared to neutral and acidic conditions. Furthermore, it has been noted that, under neutral pH conditions, SPR peak intensity increases while pH rises with a decrease under acidic pH conditions [41].

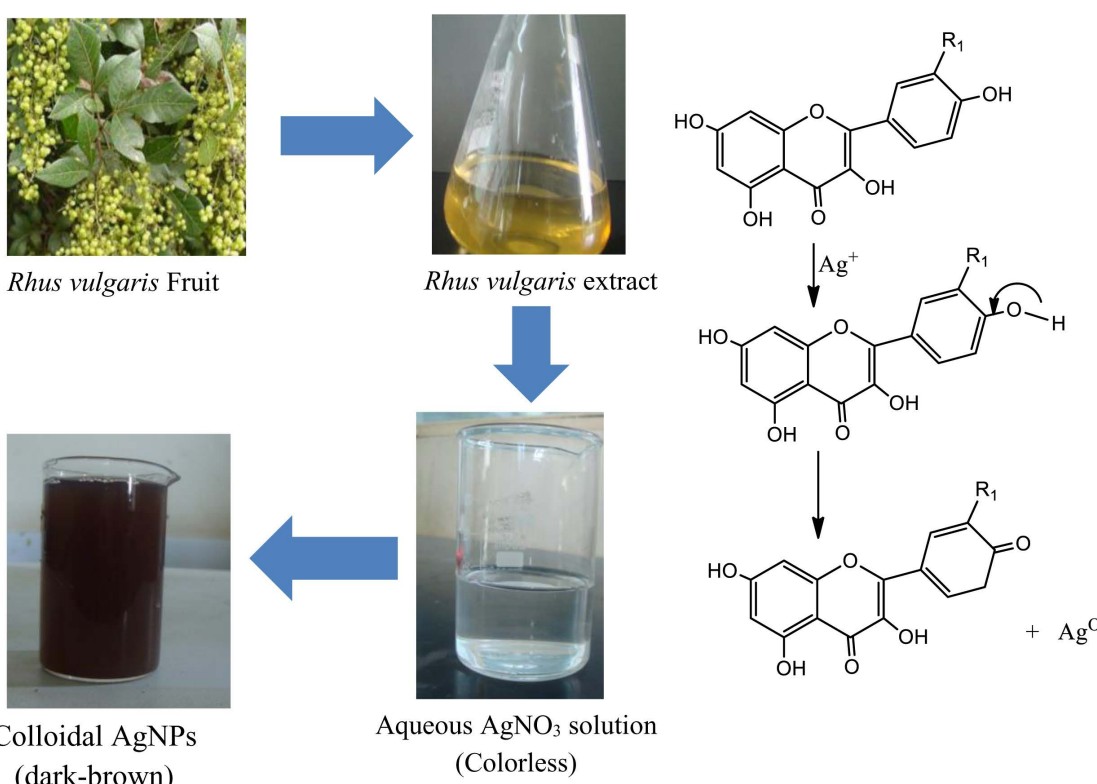

*Rhus vulgaris* Fruit

*Rhus vulgaris* extract

Colloidal AgNPs
(dark-brown)

Aqueous AgNO₃ solution
(Colorless)

**Fig 1. Green synthesis of AgNPs-RVFE and mechanism of bioactive compounds used for the synthesis.**

## Effects of temperature on the synthesis of AgNPs-RVFE

Temperature significantly affects both the nucleation and growth kinetics constant, affecting the nanoparticle's size under sufficient and insufficient $Ag_+$ precursors [42]. Table 2 illustrates how temperature affects AgNPs-RVFE production using *R. vulgaris* aqueous fruit extract based on UV-Vis absorption. While maintaining other ideal conditions constant, the spectra were recorded at 70, 80, and 90 °C. The results showed that the values were, in order, 0.179±0.01, 0.185±0.36, and 0.837±0.02. It was discovered that while the reaction mixture's maximum absorbance had dropped from 80 to 90 °C, it had increased from 70 to 80 °C. It was noted that, in contrast to 70 and 90 °C, maximum absorption was measured at 80 °C. The outcome revealed that a slightly higher temperature speed up the reduction process. As a result, utilizing *R. vulgaris* fruit extract extracted at 80 °C is the ideal condition for AgNPs synthesis, as indicated by the amount of AgNPs-RVFE produced and the UV-visible spectra of maximum absorption (Table 2). This is in good agreement with a previous report by Stavinskaya et al. [43]. They demonstrated that quick AgNPs production was seen at 80 °C along with an increase in resonant absorption intensity. The rationale is that when the reactive temperature rises, the particle size will decrease. Put differently, it is possible to produce larger-sized nanoparticles at comparatively lower temperatures. Thus, it can be said that "low temperatures promote growth" - that is, nanoparticles can grow more efficiently and larger at

**Table 2. Optimization of AgNPs-RVFE preparation using different reaction parameters (time, pH, concentration and temperature).**

| AgNPs-RVFE name | Time (minute) | pH | Conc. of EX to $AgNO_3$ ratio | Temperature (°C) | Absorbance (mean±SD) |
|---|---|---|---|---|---|
| AgRvNPs$_1$ | 40 | 4.5 | 1:9 | 80 | 0.185±0.36 |
| AgRvNPs$_2$ | 40 | 6 | 1:9 | 80 | 0.537±0.26 |
| AgRvNPs$_3$ | 40 | 7.5 | 1:9 | 80 | 0.618±0.75 |
| AgRvNPs$_4$ | 40 | 9 | 1:9 | 80 | 0.851±0.58 |
| AgRvNPs$_5$ | 30 | 9 | 1:9 | 80 | 0.762±0.02 |
| AgRvNPs$_6$ | 50 | 9 | 1:9 | 80 | 0.806±0.25 |
| AgRvNPs$_7$ | 40 | 9 | 1:9 | 70 | 0.179±0.01 |
| AgRvNPs$_8$ | 40 | 9 | 1:9 | 90 | 0.837±0.02 |
| AgRvNPs$_9$ | 40 | 9 | 3:7 | 80 | 0.702±0.25 |
| AgRvNPs$_{10}$ | 40 | 9 | 4:6 | 80 | 0.635±0.04 |
| AgRvNPs$_{11}$ | 40 | 9 | 5:5 | 80 | 0.284±0.25 |

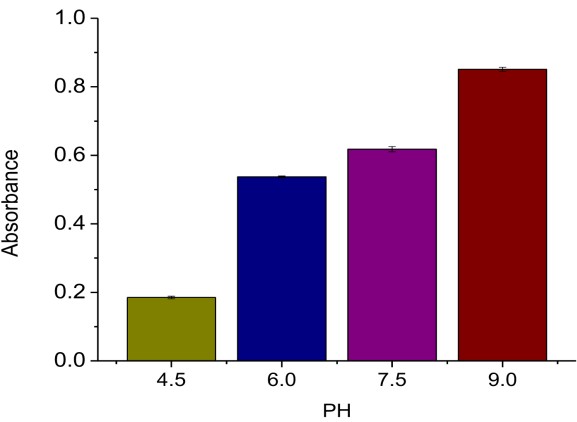

**Fig 2. Effect of pH (4.5, 6.0, 7.5, and 9.0) on green synthesis of AgNPs-RVFE.**

lower temperatures. However, it has been shown that the overall reaction rate also increases as the reaction temperature increases [42].

**Effects of reaction time for the synthesis of AgNPs-RVFE.** Studying the reaction between the aqueous plant extract and $AgNO_3$ solution for 30, 40, and 50 minutes allowed researchers to examine the impact of reaction time. As indicated in Table 2, the UV-visible measurements were taken at various time intervals. The absorption peaks at reaction times of 30, 40, and 50 minutes, keeping the other effects, such as temperature, concentration, and pH, were $0.762 \pm 0.02$, $0.851 \pm 0.58$, and $0.806 \pm 0.25$. The maximum UV-vis absorption was observed at a reaction time of 40 minutes. The smaller absorbance was observed at reaction time of 30 and 50 minutes. The study Vanaja et al. [44], confirmed the maximum absorption in the UV-vis spectrum, indicating the maximum formation of nanoparticles. The measured UV-Vis absorption spectra of the synthesized AgNPs-RVFE revealed the best peak within 420 nm at 80 min reaction time, which was consistent with the earlier research published by Sikder et al. [45] as well as Zhai et al. [46].

**Effects of silver ion concentration on the synthesis of AgNPs-RVFE.** Table 2 illustrates the impact of concentration on the synthesis of AgNPs-RVFE using fruit extract from *R. vulgaris*. The plant extract to silver ion solution ratio was observed at different concentrations for this operational parameter. *R. vulgaris* fruit extract was used in the experiment at ratios of 3:7, 4:6, 5:5, and 1:9, respectively, to silver ion concentration. As can be shown in Table 2, the intensity rises as the concentration of $Ag_+$ ions rises while holding the other variable constant. At 420 nm in wavelength and 1:9 concentration, the highest absorbance was measured at $0.179 \pm 0.01$. However, the smallest absorbance was observed at a concentration of 5:5 ratios. The light brown color of the nanoparticles was observed with the 3:7 and 4:6 ratios of plant extract to $AgNO_3$ solution. Darker colour shades were observed at 1:9 ratios. This showed an optimal value for the synthesis of AgNPs-RVFE was observed at ratio of 1:9 $Ag_+$ concentrations to *R. vulgaris* fruit extract. This result confirmed that the highest concentration of $AgNO_3$ resulted in the strongest UV absorption, indicating the highest amount of synthesized AgNPs-RVFE.

## Characterization of synthesized silver nanoparticle

**UV–Vis spectra analysis of AgNPs-RVFE.** The biosynthesized AgNPs-RVFE were first characterized using the ultraviolet-visible (UV–Vis) spectrum, which is the simplest and least complex technique that can reveal the formation of metal nanoparticles—as long as the metal nanoparticles show surface plasmon resonance (SPR). The free mobility of electrons in AgNPs-RVFE' valence and conduction bands is what gives rise to the SPR absorption band [47]. AgNPs-RVFE production was validated by color changes in the extracts. The production of AgNPs-RVFE was validated by spectrophotometric measurement of the colored solution using a Shimadzu UV-vis spectrophotometer operating in the 280–800 nm spectrum range. This study revealed an SPR increase between 410 and 420 nm (Fig 3). These results fully agree with the literature analysis of Zhang et al. [48] and Nyoni and Muzenda.[49] They demonstrate how well plant materials work in the phytosynthesis of nanomaterials from silver ions in solution.

When compared to the other AgNPs-RVFE samples, it was found that the maximum absorption was slightly higher in the $AgRvNPs_4$ sample, which had a maximum wavelength of 415 nm (Fig 3). This indicates that appropriate AgNPs-RVFE were produced at 80 °C, pH of 9, reaction period of 40 minutes, and concentration of 1:9 (RVE:$AgNO_3$ solution). There was a bathochromic shift in the plasmon peak from AgRvNPs7 to AgRvNPs4. The bathochromic shift indicated a decrease in the mean diameter of $AgRvNPs_4$ [50]. Sample $AgRvNPs_{10}$ had the lowest UV absorbance of all the samples. This variation in absorbance intensity is caused by the plant extract's varying ability to convert $Ag_+$ to $Ag^0$. After being mixed with the plant extract, the silver ions within the extract were converted to AgNPs-RVFE. The plant extracts changed into a variety of colors, including pale yellow, reddish brown, light yellow and dark amber. This color shift demonstrated how silver nitrates were reduced to AgNPs, which depends on the bioactive substances in the sample [32].

**Fourier transform infrared analysis.** The functional groups used as the capping and reduction processes were identified using the information obtained from the FTIR measurements. Fig 4 displays the FT-IR absorption spectrum analysis of green AgNPs-RVFE utilizing RVE in the 400–4000 $cm^{-1}$ range.

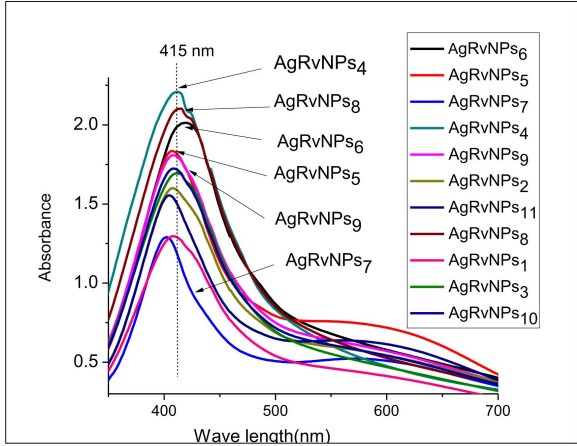

**Fig 3. UV–Vis absorption spectra of AgNPs-RVFE synthesized by R. vulgaris at different AgNO3 concentrations.**

Plant RVE contains alkaloids, saponins, terpenoids, tannin, flavonoids, glycosidase, and phenol—all of which are essential for synthesizing AgNPs-RVFE. A few distinct compounds decrease Ag ions to their equivalent NPs. FT-IR was used to describe the synthesis of AgNPs using *R. vulgaris*. As observed in FT-IR peaks (Fig 4), AgRvNPs$_4$ samples possess different functional group. C=O stretching is observed at 1650 cm_1 and OH-bond stretching found at 3330 and 3310 cm_1. The FT-IR peak indicated a broadening band at 650, 1650, and 3330 cm_1 after reducing Ag$_+$ ions, indicating the importance of functional groups such as C-H, C=O, and OH− respectively for the synthesis of AgNPs [51]. The stretching vibration between 1650 cm$^{-1}$ and 1366 cm$^{-1}$ indicates the presence of a carbon double bond in benzene [52]. The presence of Ag-O bond is indicated by peaks of absorption bands at 1366 cm-1 and 650 cm$^{-1}$ in the FT-IR spectrum of AgRvNPs$_4$ [53], This demonstrates that plant extracts are used to synthesize AgNPs.

**Zeta potential.** More stable nanoparticles are produced by stronger electrostatic repulsion between those with larger zeta potential values. The optimized AgNPs i.e AgRVNPs$_4$ zeta potential's dramatic peak, shown in Fig 5, was determined to be -26.0 mV, confirming the stability of the formulation [54]. In the work of Ghazwani et al. [55], approximately -25.0 mV was found in AgNPs synthesis using *Methyl anthranilate*.

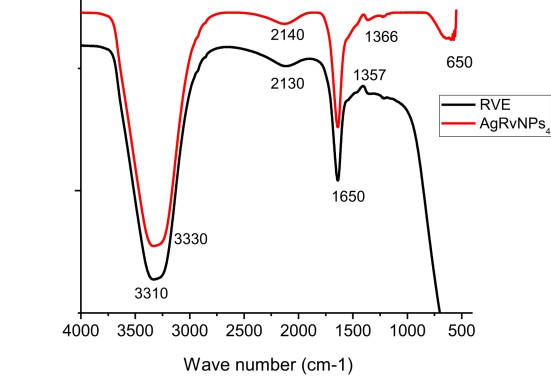

**Fig 4. FTIR spectra of** (a) RVE and (b) Synthesized AgRVNPs4.

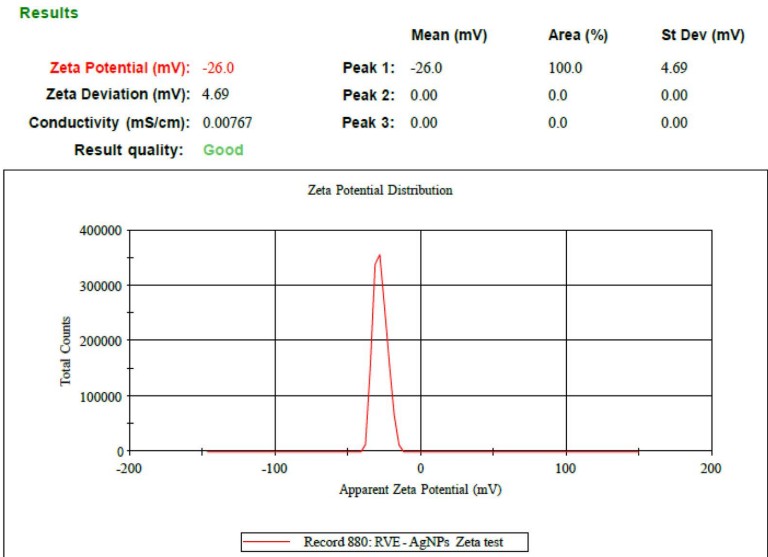

**Results**

|  |  | Mean (mV) | Area (%) | St Dev (mV) |
|---|---|---|---|---|
| Zeta Potential (mV): -26.0 | Peak 1: -26.0 | | 100.0 | 4.69 |
| Zeta Deviation (mV): 4.69 | Peak 2: 0.00 | | 0.0 | 0.00 |
| Conductivity (mS/cm): 0.00767 | Peak 3: 0.00 | | 0.0 | 0.00 |
| Result quality: Good | | | | |

**Fig 5. Zeta potential of optimized silver nanoparticle (AgRvNPs4).**

## X-ray diffraction (XRD) analysis

The crystalline structure and lattice properties of the produced AgNPs were examined using the XRD method. The Cu Kα, whose wavelength range is 1.5406 A°, is used to calculate angle 2θ values between 20° and 100°. The XRD analysis of the optimized silver nanoparticle (AgRvNPs$_4$) is observed in Fig 6. As indicated in Fig 6, four distinct peaks are formed at 2θ values of 38.4°, 44.5°, 64.90°, and 77.4°. The diffraction planes (111), (200), (220), and (311) of the FCC structure of the artificially synthesized AgNPs are represented by these 2θ values. The optimal orientation of the structure is shown by the greatest intensity peak located at the (111) plane. The bioactive compounds in the *R. vulgaris* fruit extract

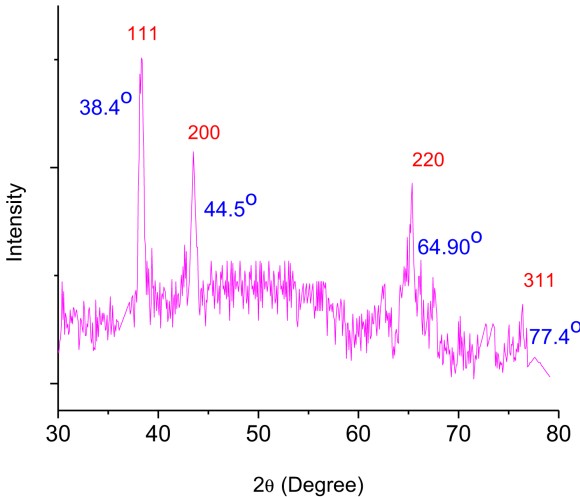

**Fig 6. XRD pattern of optimized silver nanoparticle (AgRvNPs4).**

are responsible for some of the extra star peaks seen in the graph. Fig 6 also shows the crystallite size (D) of AgRvNPs4 using equation 3 by the Debye-Scherrer formula. The result confirmed that the crystallite size is 10.00 nm.

$$Crystalline\ size\ (D) = \frac{k \times \lambda}{\beta \times cos\theta}$$

(3)

Where λ is the X-ray radiation wavelength (1.54178Å), θ is the scattering angle in radians, D is the crystallite size in Å, k is Scherer's constant (k = 0.94), and β is the integral breadth of the maximum complete width (FWHM).

In the present study, R. vulgaris was used to generate AgNPs, which produced spherical, 6–24 nm-sized nanoparticles. Green synthesis techniques for nanoparticles, which use natural plant extracts as stabilizing and reducing agents, are becoming more and more popular.

AgNPs made from *R. vulgaris* (6–24 nm) are noticeably smaller than those made from several other plant extracts, as shown in Table 3. In contrast to smaller nanoparticles, *A. lanata* produced larger nanoparticles (95 nm), which would have limited their surface area and reactivity. The nanoparticles produced by *J. curcas* (seed) were larger than those from *R. vulgaris* but remained relatively small, measuring between 15 and 20 nm. A wider size range (5–50 nm) was displayed by *V. cinerea*, suggesting particle size heterogeneity that may impact application consistency. *V. sinaiticum* produced nanoparticles with sizes ranging from 2 to 40 nm; the lower end was comparable to the current study, suggesting that similar applications may be possible. Furthermore, compared to *R. vulgaris*, *T. grandis* (10–30 nm) and *A. indica* (20–30 nm) nanoparticles were larger.

AgNPs from *R. vulgaris* have a spherical shape, which is consistent with most previous research that reports spherical nanoparticles. Because of its consistency and stability, this shape is frequently chosen because it improves nanoparticle performance in various applications. In contrast to the uniform spherical shape seen in the current work, *V. cinerea* is known to exhibit a polydisperse size distribution, which could result in inconsistent behavior and applicability.

Since the AgNPs made from *R. vulgaris* are some of the smallest ones included in this table, their reactivity and their uses in industries like environmental remediation, medicine, and catalysis may be increased. In specific applications, like as medication delivery systems or antibacterial agents, where smaller nanoparticles are preferred, the size and form of AgNPs derived from *R. vulgaris* may offer benefits. The production of AgNPs from *V. sinaiticum* (2–40 nm) suggests the possibility of creating extremely tiny nanoparticles, which might have special qualities but could also pose stability and aggregation issues.

### Transition electron microscope (TEM)

TEM provides high-resolution pictures and analyses the internal structure of materials at the atomic or nanometer scale [65]. Fig 7 depicts a typical transmission electron microscopy image of the produced *R.vulgaris* fruit extract-mediated

**Table 3. Synthesis of AgNPs from Various Plant Extracts: Size, Shape, and Plant Parts Used.**

| Plant extract | Part of the plant | Size (nm) | Shape | References |
|---|---|---|---|---|
| *A. indica* | Leaf | 20-30 | spherical | [56] |
| *A. lanata* | Whole plant | 95 | sphrical | [57] |
| *B. sensitivum* | Whole plant | 8 | spherical | [58] |
| *J. curcas* | Seed | 15-20 | spherical | [59] |
| *P. guajava* | Leaf | 18-20 | spherical | [60] |
| *R. vulgaris* | fruit | 6–24 | spherical | Current work |
| *T. bellirica* | Fruit | 20.6 | spherical | [61] |
| *T. grandis* | seed | 10-30 | spherical | [62] |
| *V. cinerea* | Whole plant | 5-50 | Polydisperse | [63] |
| *V.sinaiticum* | leaf | 2–40 | spherical shape | [64] |

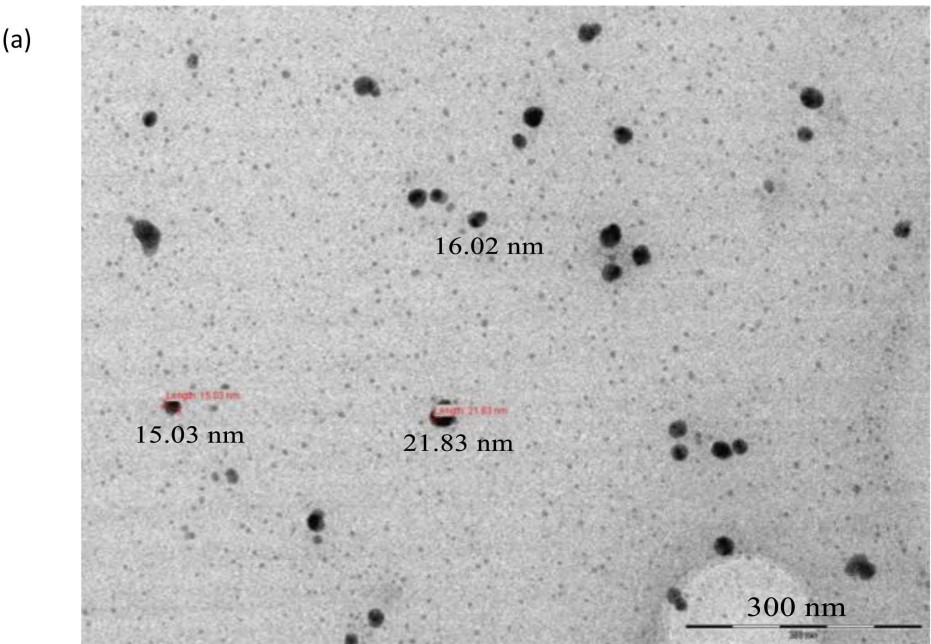

(a)

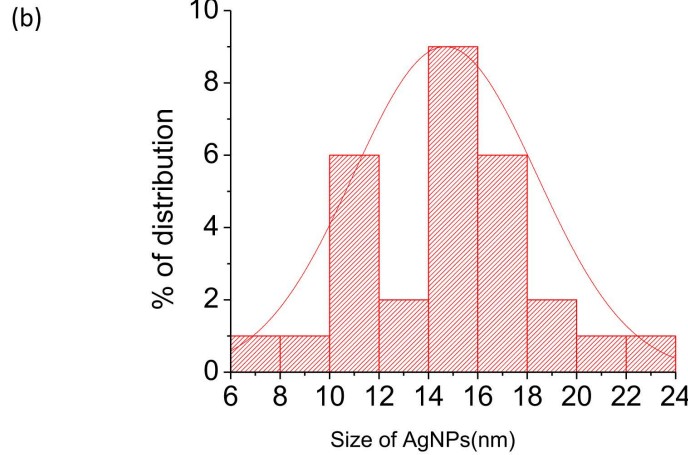

(b)

Fig 7. **TEM image of optimized silver nanoparticle (AgRvNPs4)** (a) and its percentage of distribution (b).

silver nanoparticle (AgRvNPs4). It is clear that the nanoparticles were widely distributed and primarily spherical in shape with a range of 6–24 nm for particle size. The size of the nanoparticle was measured using ImageJ software scanning from TEM images (Fig 7). The average value of nanoparticle size is about (14.64 nm). That is in agreement with the previous studies done by Siddiqui et al. [66]. Their finding showed that the produced AgNPs had an average size of 15 nm and ranged in size from 10 to 20 nm. In another work by Kalpana et al. [67], the TEM image showed that the AgNPs made from *T. nucifera's* aqueous extract possess a nanoparticle size in the range between 10 and 125 nm with a spherically homogeneous shape. In addition to the above experimental work, Serra et al. [68] produced well-dispersed spherical-shaped AgNPs, whose sizes ranged between 1.5 and 13 nm and had a monodisperse size distribution.

## Test sun protection activities

The SPF is a number that represents a sunscreen product's effectiveness. A sunscreen lotion's ability to absorb light between 290 and 400 nm helps prevent sunburn and other skin damage [69]. For a very long time, in vivo tests carried out on human volunteers have been used to evaluate the effectiveness of sunscreen formulations. Aside from the fact that in vivo tests take much time and are typically unpredictable, there are also moral concerns with using humans for testing [69]. In vitro SPF is helpful for screening tests during product development and in vivo SPF measurements.

Because of its special qualities, such as its antibacterial activity, biocompatibility, and capacity for wound healing, AgNPs have attracted much interest for their sun protection capabilities [70]. AgNPs have strong antibacterial properties against various pathogens, such as viruses, fungi, and bacteria. Because of this feature, they are especially useful in dermatological applications, such as treating skin infections and avoiding wound biofilm formation [71]. The ability of AgNPs to release silver ions in a controlled manner contributes to their therapeutic effects, making them suitable for use in topical formulations and dressings. Studies have demonstrated that AgNPs can accelerate tissue regeneration, reduce inflammation, and enhance cellular migration and proliferation to promote wound healing [72]. AgNPs are used more often in cosmetic goods due to their antibacterial qualities and ability to enhance skin appearance and medicinal applications. Their application in formulas for skin regeneration, anti-aging, and acne therapy is gaining popularity [73].

This study examined three commercial sunscreen products: yellow-colored Vaseline lotion (YVL), Aloe Vera Vaseline lotion (AVL), and Rounsun honey lotion (RHL). Three other samples containing a mixture of $AgRvNPs_4$ and YVL ($AgRvNPs_4$-YVL), $AgRvNPs_4$ and AVL ($AgRvNPs_4$-AVL), and $AgRvNPs_4$ and RHL ($AgRvNPs_4$-RHL) also evaluated by UV spectrophotometry applying Mansur mathematical equation [69]. The SPF labeled values fell between 21.1 and 234.5 (Table 4). As observed in Table 4, it is evident that the SPF values for the AVL, RHL, and YVL samples closely match the SPF that was labeled in each lotion. However, the SPF value of AgNPs containing samples such as AgRvNPs4-AVL, $AgRvNPs_4$-RHL, and AgRvNPs4-YVL samples was higher than the amount indicated on the label of the samples.

Table 4 shows that $AgRvNPs_4$-AVL has a maximum absorbance that is higher than all samples with known sunscreen concentrations. This is most likely because sample $AgRvNPs_4$-AVL contains more sunscreen material overall than the other samples. Sample AVL has fewer sunscreens overall (SPF = 21.1) than sample $AgRvNPs_4$-AVL (SPF = 234.5); Sample RHL contains fewer sunscreens overall (SPF = 30.7) than sample $AgRvNPs_4$-RHL (SPF = 88.3); and Sample YVL contains fewer sunscreens overall (SPF = 64.0) than sample $AgRvNPs_4$-YVL (SPF = 176.0) (Fig 8). It was discovered that adding $AgRvNPs_4$ to AVL, RHL, and YVL increased their sun protection factor by 2.70, 11.16, and 2.90 times, respectively. This might be due to loading $AgRvNPs_4$ into the aforementioned three samples, which could enhance UV absorption through the interaction between the materials and the plasmon resonance of $AgRvNPs_4$ [74]. The other explanation could be that $AgRvNPs_4$ have a more significant surface area, which could improve the active ingredient's dispersion and increase UV radiation protection [75]. The two main characteristics of $AgRvNPs_4$ that make it excellent for sun screening are its particle

**Table 4. The absorbance of each sample between (290- 320 nm).**

| Wave length (nm) | Absorbance's | | | | | |
|---|---|---|---|---|---|---|
| | RLH | YVL | AVL | $AgRvNPs_4$-RHL | $AgRvNPs_4$-AVL | $AgRvNPs_4$-YVL |
| 290 | 0.50800 | 0.69167 | 0.44233 | 0.77700 | 1.21033 | 1.07100 |
| 295 | 0.45967 | 0.60300 | 0.36367 | 0.70633 | 1.11767 | 0.99967 |
| 300 | 0.42233 | 0.57567 | 0.34100 | 0.68600 | 1.11267 | 0.97067 |
| 305 | 0.41000 | 0.54767 | 0.31400 | 0.65600 | 1.04600 | 0.96400 |
| 310 | 0.39900 | 0.53667 | 0.29700 | 0.62100 | 0.99833 | 0.87667 |
| 315 | 0.33767 | 0.48233 | 0.24100 | 0.56667 | 0.95167 | 0.77800 |
| 320 | 0.27167 | 0.3800 | 0.14633 | 0.49367 | 0.94233 | 0.70500 |

size and band gap [76]. A larger band gap and smaller particle size work better for a UV absorber. Synthesized nanoparticles may be a preferable option for use in sunscreens since, based on prior research, Ag nanoparticles have a bigger band gap [77] and their size is 14 nm.

## Antioxidant activity of AgNPs-RVFE

In order to lessen oxidative damage, antioxidants give free radicals electrons and conceal their detrimental effects from biological processes [78]. The antioxidant activity of the particles was ascertained by measuring the reduction in DPPH absorbance at 517 nm wavelength and evaluating the ability of the green-synthesised AgNPs to scavenge free radicals generated by DPPH [79]. Antioxidant substances can decrease the violet-colored DPPH radical to DPPH-H (yellow hue) by removing hydrogen atoms or electrons from the radical. Spectrophotometry is used to track the color change of the DPPH in order to assess the scavenging activity of the generated AgNPs. The blue hue turning yellow indicated that the nanoparticles had antioxidant properties [80]. The ability of different natural or phytochemicals to scavenge free radicals is commonly studied using DPPH radicals as a model system [81].

The antioxidant activity of optimized *R. vulgaris* fruit extract-mediated silver nanoparticles such as AgRvNPs$_4$ and ascorbic acid (AA) is shown in Fig 9. The results show that the antioxidant experiment conducted on nanoparticles reveals significant antioxidant capacity. The AgRvNPs$_4$ DPPH activity increased in a concentration-dependent manner. The AgRvNPs$_4$ concentrations that scavenged the DPPH radical rose from 20 to 100 μg/ml (Fig 9b). The percentage of DPPH radical inhibition ranged from 13.00% ± 1.1 (20 μg/ml concentration) to 53.7% ± 0.12 (100 μg/ml concentration). For AA scavenging activities at 100 μg/ml concentration, a higher value of 85.71% ± 0.51 was observed compared to AgRvNPs$_4$, synthesized by *R. vulgaris* fruit extract. However, the scavenging activities of AgRvNPs$_4$ have the highest antioxidant potential (84.9 ± 0.6) than the other samples. As observed in the error bar in Fig 9b, there was no significant difference ($P > 0.05$) (in DPPH scavenging activities between AA and AgRvNPs$_4$ samples.

The absorbance at 517 nm changed when AgRvNPs$_4$ samples were added to the DPPH solution as observed in Fig 9a. The maximum intensity at 517 nm gradually reduced over time in agreement with variations in the color of the DPPH solution in the presence of AgRvNPs$_4$. Since AgRvNPs$_4$ scavenges DPPH by donating a hydrogen atom to stabilize the DPPH molecule, the color intensity decline demonstrated that AgRvNPs$_4$ could scavenge free radicals [82]. AA, RVE, and AgRvNPs$_4$ all successfully inhibited DPPH, with IC$_{50}$ values ranging from 69 to 81 μg/ml. Notably, AgRvNPs$_4$ was found to have an IC$_{50}$ value of 81.2 μg/ml, indicating its effectiveness in scavenging free radicals. In contrast, RVE demonstrated

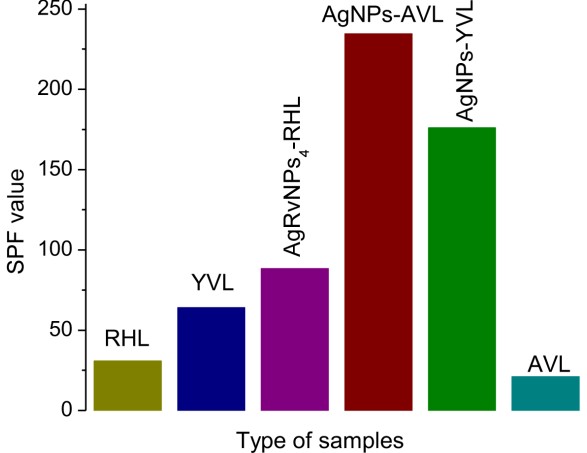

**Fig 8. The SPF value of different samples of AgRvNPs4 mediated by RVE.**

(a)

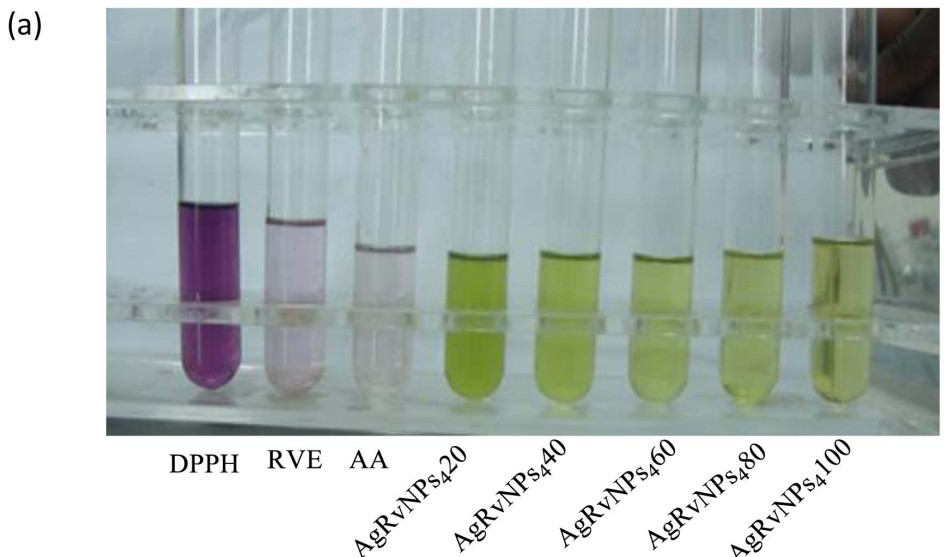

(b)

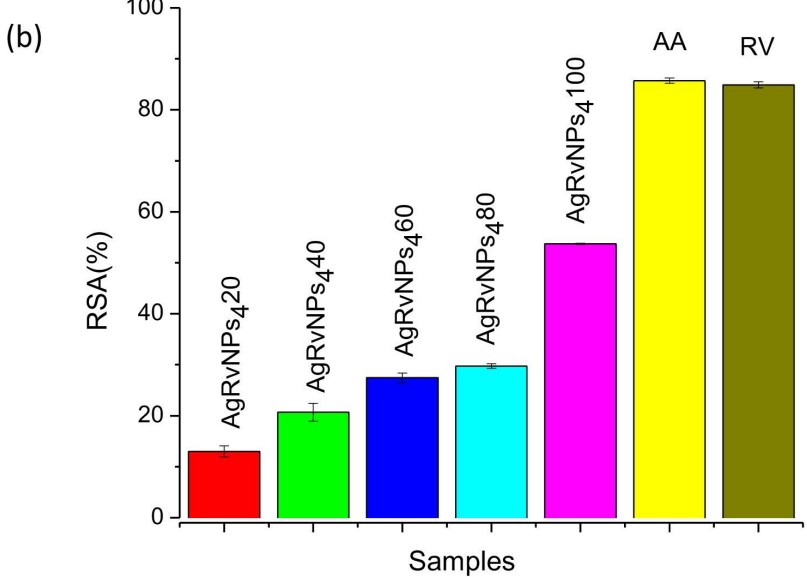

**Fig 9. Antioxidant power of AgRvNP4 samples** (a) with color change and (b) RSA value.

a slightly stronger inhibitory effect, with an IC$_{50}$ value of 69.1 µg/ml. The value of IC$_{50}$ for standard AA was 79.7 µg/ml. The different levels of antioxidant activity among these substances are demonstrated by this comparison, which implies that although all three have the ability to block DPPH, RVE is the most effective. This showed that RVE demonstrated the highest level of antioxidant activity compared to other samples, such as AA and AgRvNPs$_4$, because the lower the IC$_{50}$ values, the higher the antioxidant activities. The difference in the value of IC$_{50}$ occurred due to the difference in the size

between them. The difference in antioxidant activities between these samples is due to the difference in their size between them. Numerous investigations have demonstrated that the existence of different OH groups in flavonoids and phenolic bioactive compounds is what causes metal oxide nanoparticles to develop and stabilize [83]. Additionally, the phytochemicals in *R. vulgaris* are capped over AgRvNPs$_4$, can donate hydrogen, and give the nanoparticles a strong antioxidant potential [84].

Several researchers reported similar results of AgNPs by DPPH radical scavenging; for example, in the work of Melkamu and Bitew [40], the capacity to scavenge DPPH radicals improved when the concentration of green-produced AgNPs mediated by *Hagenia abyssinica* rose between 10 µg/mL and 320 µg/mL. The scavenging activity showed a dose-dependent pattern that rose as the concentration of AgNPs increased. In comparison to regular AA, the green-produced AgNPs had a decreased scavenging potential. The strong radical scavenging activity of AgNPs mediated by *E. abyssinica* aerial parts was demonstrated by Mukaratirwa et al. [85]. The results showed that the AgNPs' antioxidant properties were comparable to those of AA (90.1±0.10%), with 73.4% scavenging activity at 400 µg/mL. An investigation was conducted to evaluate the effect of plant extract on the antioxidant properties of green-synthesized nanoparticles. The results revealed that AgNPs exhibited superior inhibition of DPPH free radicals (80.78%) at a concentration of 40 µg/mL compared to the plant extract alone [86]. In the work of Das et al. [87] comparative study on antioxidant properties of bio-synthesized AgNPs using outer peels of two varieties of *Ipomoea batatas* such as Korean red skin sweet potato (Ib1) and Korean pumpkin sweet potato (Ib2) was conducted. The result showed that the antioxidant activities of AgNPs mediated by potato samples are higher than that of potato samples only. However, the antioxidant prospective was higher in Ib2-AgNPs than in Ib1-AgNPs; this is due to Ib2-AgNPs exhibiting relatively higher functional activity than Ib1-AgNPs.

**Antibacterial activity** The antibacterial activity of biosynthesized AgNPs derived from RVE was evaluated using the standard agar well diffusion method. This study involved comparing the synthesized nanoparticles against clinically isolated strains of four different microbial species. Ciprofloxacin was utilized as a positive control, allowing for a benchmark measurement of the test solution's effectiveness by assessing the diameter of the resulting zone of inhibition (ZOI). Fig 10 and Table 5 demonstrate the antibacterial activity of each tested sample against gram-positive (*S. aureus* and *S. epidermidis*) and gram-negative (*E. coli* and *K. pneumoniae*) pathogens. The value indicated that the optimized silver nanoparticle (AgRvNPs$_4$) showed high antibacterial activities. This antibacterial efficacy may be associated with their tiny size, dispersion, and spherical morphology, which provide a broad surface for maximum interaction with bacteria and cause even more damage than bigger particle sizes [88]. As observed in Fig 5, the lowest zone of inhibition was demonstrated by AgRvNPs$_4$25 against *E. Coli* (30.21±1.21 mm), *K. pneumonia* (29.54±1.11 mm), *S. aureus* (23.88±1.10 mm), and *S. epidermidis* (29.10±1.52 mm). As listed in Table 5, the AgRvNPs$_4$75 sample had the maximum zone of inhibition (40.54±1.51 mm) against the gram-negative bacteria species *K. pneumonia*, while the AgRvNPs$_4$25 sample had the lowest antibacterial activity (23.88±1.10 mm) against *S. aureus*. All bacteria samples have increased antibacterial activity in the following order: AgRvNPs$_4$25,

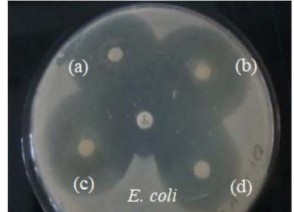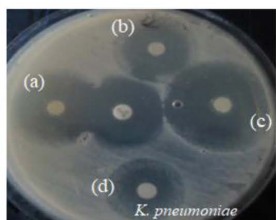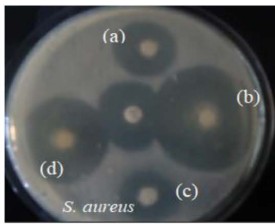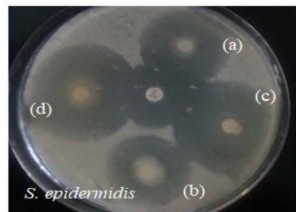

**Fig 10. The zone of inhibition exhibited by AgRvNPs4 against four bacteria species.** (A) E. coli, (B) K. pneumonia, (C) S. aureus, and (D) K. pneumonia and (a) AgRvNPs425, (b) AgRvNPs425, (c) AgRvNPs475, and (d) AgRvNPs4100.

**Table 5. The zone of inhibition that AgNPs-RVFE against _K. pneumonia, S. aureus, S. epidermidis,_ and _E. coli._**

| Bacteria species | Samples | | | | Cipro- floxacin |
|---|---|---|---|---|---|
| | AgRvNPs₄25 | AgRvNPs₄25 | AgRvNPs₄75 | AgRvNPs₄100 | |
| _E. coli_ | 30.21 ± 1.21[a] | 38.88 ± 1.20[b] | 38.54 ± 0.95[b] | 39.21 ± 1.05[b] | 28.00 ± 00 |
| _K. pneumonia_ | 29.54 ± 1.11[a] | 37.54 ± 1.02[b] | 40.54 ± 1.51[b] | 38.88 ± 1.00[b] | 28.00 ± 00 |
| _S. aureus_ | 23.88 ± 1.10[a] | 30.21 ± 2.01[b] | 31.21 ± 1.00[b] | 38.21 ± 1.08[c] | 28.00 ± 00 |
| _S. epidermidis_ | 29.10 ± 1.52[a] | 28.54 ± 1.15[a] | 34.88 ± 1.13[b] | 36.54 ± 1.15[b] | 28.00 ± 00 |

The mean value ± SD was utilized after more than three examinations of each data set. Subscriptions a and b have values in the same column that differ significantly (P ≤ 0.05).

AgRvNPs₄25, AgRvNPs₄75, and AgRvNPs₄100. For example, the ZOI values for _S. epidermidis_ bacteria were, for AgRvNPs₄25, AgRvNPs₄50, AgRvNPs₄75, and AgRvNPs₄100, respectively, 29.10 ± 1.52, 28.54 ± 1.15, 34.88 ± 1.13, and 36.54 ± 1.15. The ZOI value of the common antibiotic Ciprofloxacin is 28 ± 0.00, practically the lowest compared to most bacterial species' ZOI values. Based on the results above, it can be said that green synthesis of AgNPs with _R. vulgaris_ fruit extract has strong antibacterial potential. According to the antibacterial activity test results, gram-negative bacterial strains had a higher ZOI value than Gram-positive bacterial strains.

The strongest antibacterial activity of green-generated AgNPs-RVFE against Gram-negative bacteria was probably caused by their wide surface area, which allows for better interaction with the microorganisms and will have a bigger bactericidal impact than larger particles [89]. Gram-positive bacteria contain many layers of peptidoglycan in their membrane, making them more stiff than gram-negative bacteria with a single layer. Their varying cell wall compositions could contribute to this antibacterial variation. When silver ions from nanoparticles sense an electrostatic attraction from the bacterial cell wall, they migrate and adhere to it due to the wall's negative charge. As a result, the cell wall's permeability and composition are altered [71].

The antibacterial properties of AgNPs mediated by medicinal herbs were noted in several studies. Asif et al. [90] demonstrated the antibacterial properties of AgNPs colloidal samples in vitro using varying AgNPs concentrations (100.0, 50.0, and 25.0 μg/ml) against Gram negative _E. coli_ using the well diffusion method. The result indicated that there is a dose-dependent increase in antibacterial activity. An increased level of antibacterial activity was noted at a 100 μg/ml concentration. Against every studied bacterial strain, Melkamu and Bitew [40] synthesized AgNPs and plant leave extract showed antibacterial properties. However, both the plant crude extract and the manufactured nanoparticle worked better against Gram-negative bacteria species such as _S. typhimurium_. The concentration of the generated AgNPs and plant leaf extract increased in correlation with the antibacterial activity. Green-produced AgNPs antibacterial activity against gram-negative bacteria such as _S. typhimurium_ peaked at a concentration of 200 μg/mL, with a ZOI of 18.3 mm. Additionally, Mukaratirwa et al. [85] demonstrated that at a dosage of 100 μg/mL of nanoparticles, the zones of inhibition for _E. Coli_ and Salmonella were determined to be 16.9 ± 0.10 and 17.8 ± 0.11 mm respectively. The biggest inhibitory zone was measured with _E. coli_ at 17.8 ± 0.11 mm at 100 μg/mL of AgNPs. The diameter of the limitation area for developing _B. subtilis, B. vallismortis,_ and _E. coli_ bacteria was measured to be 15, 16, and 12, respectively, with distilled water acting as a control. In contrast, Gram-positive bacteria are more vulnerable to the antibacterial activity of the biosynthesized AgNPs than Gram-negative bacteria, according to the findings of Pirtarighat et al. [91] Using _Tabebuia rosea_ seeds as a reducing agent, the study synthesized AgNPs and verified their antibacterial effectiveness against both Gram-positive and Gram-negative bacteria. At the highest dose of 100 mg ml₋¹, _E. faecalis, P. aeruginosa, P. mirabilis,_ and _S. aureus_ were the most inhibited, with _K. pneumonia_ showing the largest inhibition zone of 20 ± 0.48 mm [92].

## Conclusions

The study successfully demonstrated the use of *R. vulgaris* fruit extract as a stabilizing and reducing agent in producing silver nanoparticles (AgNPs). The synthesized AgNPs exhibited notable biological activities, including strong antioxidant and antibacterial properties, as well as effective UV protection. Characterization techniques such as UV-Vis spectroscopy, FT-IR, TEM, XRD, and zeta potential analysis confirmed the successful synthesis and stability of the nanoparticles, which had an average size of approximately 14.64 nm and zeta potential of -26.0 mV. The antioxidant activity of AgNPs-RVFE was significant, with a maximum DPPH radical scavenging inhibition of 53.7% at a concentration of 100 µg/mL. Anti-bacterial assays revealed that AgNPs-RVFE exhibited substantial inhibition against various bacterial strains, particularly Gram-negative bacteria, with inhibition zones ranging from 23.88 mm to 30.21 mm. Additionally, the synthesized AgNPs demonstrated a high Sun Protection Factor (SPF) of 234.5, indicating their potential as effective UV blockers. These findings highlight the potential of *R. vulgaris* fruit extract in the green synthesis of AgNPs, which have promising applications in nanomedicine, biotechnology, and healthcare. The eco-friendly nature of this synthesis method, combined with the significant biological activities of the resulting nanoparticles, suggests that AgNPs-RVFE could be valuable in developing new therapeutic agents and formulations for various applications. Further research is warranted to explore the full range of applications and mechanisms of action of these nanoparticles.

## Acknowledgments

We thank Debre Tabor and Addis Ababa Science and Technology University for supporting our laboratory work and for providing reliable lab instruments to characterize the results.

## Author contributions

**Investigation:** Woinshet Kassie Alemu, Limenew Abate Worku.

**Methodology:** Woinshet Kassie Alemu, Limenew Abate Worku.

**Software:** Archana Bachheti.

**Supervision:** Rakesh Kumar Bachheti.

**Writing – original draft:** Woinshet Kassie Alemu, Limenew Abate Worku.

**Writing – review & editing:** Limenew Abate Worku.

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
