## [Decision Letter · Decision Letter 0]

19 Jan 2025

PONE-D-24-53986Rhus vulgaris Fruit-Mediated Silver Nanoparticles: Synthesis, Characterization, and Potential for Sun Protection, Antioxidant and Antibacterial ApplicationsPLOS ONE

Dear Dr. Abate,

Thank you for submitting your manuscript to PLOS ONE. After careful consideration, we feel that it has merit but does not fully meet PLOS ONE’s publication criteria as it currently stands. Therefore, we invite you to submit a revised version of the manuscript that addresses the points raised during the review process.

We look forward to receiving your revised manuscript.

Kind regards,

Chellasamy Panneerselvam

Academic Editor

PLOS ONE

Journal Requirements:

Reviewers' comments:

Reviewer's Responses to Questions

**Comments to the Author**

1. Is the manuscript technically sound, and do the data support the conclusions?

Reviewer #1: Yes

Reviewer #2: Partly

2. Has the statistical analysis been performed appropriately and rigorously? 

Reviewer #1: Yes

Reviewer #2: Yes

3. Have the authors made all data underlying the findings in their manuscript fully available?

Reviewer #1: Yes

Reviewer #2: No

4. Is the manuscript presented in an intelligible fashion and written in standard English?

Reviewer #1: Yes

Reviewer #2: No

5. Review Comments to the Author

Reviewer #1: Dear Author, the manuscript entitled Rhus vulgaris Fruit-Mediated Silver Nanoparticles: Synthesis, Characterization, and

Potential for Sun Protection, Antioxidant and Antibacterial Applications was written good but still needs to be revised with the following points.

1. The abstract can be modified and rewritten.

2. The significance of R. vulgaris can be focused in the first part of the introduction.

3. The exploration of AgNPs can be effectively written in the introduction with specific biological properties.

4. Subheadings plant collection and preparation of plant extracts can be merged together and write in simplified manner.

5. Remove the subheadings of characterization part and merge together.

6. Why has the author done only one antioxidant activity?

7. The optimization part can be emphasized more with some more factors.

8. The discussion related with AgNPs towards skin related research can be discussed more.

9. Check for overall grammatical errors and sentence framing.

Reviewer #2: Authors presented “Rhus vulgaris Fruit-Mediated Silver Nanoparticles: Synthesis, Characterization, and Potential for Sun Protection, Antioxidant and Antibacterial Applications”, however, it requires more characterization part.

Authors should include and explain the below comments in the revised manuscript

1. Abstract should be rewrite, and should summarize the reason for the work, the most significant results, and the conclusions.

2. In introduction part, several categories of metallic NPs by green synthesis such as Au, Ag, Zno, Pd, TiO2, Cu, etc, have been reported with improved biomimetic attributes. Include these metal and metal oxide NPs with reference, Au (Phytosynthesis of gold nanoparticles using Caesalpinia pulcherrima (peacock flower) flower extract and evaluation of their antimicrobial activities) Ag (https://doi.org/10.1016/j.matpr.2020.10.048), ZnO and Cu (https://doi.org/10.1007/s43630-022-00224-0), Fe and Pd (DOI: 10.1039/C5CY00099H), etc.

3. Synthetic part: Why the authors used Rhus vulgaris Fruit extract only instead of other extract?

4. So many reports already published using the Rhus vulgaris Fruit extracts for the synthesis of AgNPs and biomedical application studies?

5. UV and IR spectral image should be clear, IR is not clear better to remove it

6. Characterization part: all figures including IR, TEM, should be clear

7. XRD for AgNPs required with clear hkl values (111, 200, 220, 311, 222 for AgNPs) and explain the hkl values

8. Single particle or HRTEM should be included with EDAX

9. Author should explain how the average size of AgNPs is 10 nm to 20 nm?

10. In addition, the lattice fringe spacings or D spacing hkl should be included

11. For antioxidant activity: author performed DPPH, what about ABTS, and NOx radical scavenging analyses? Compare and refer (https://doi.org/10.2147/IJN.S210517)

12. For antibacterial activity refer (DOI 10.1088/2053-1591/ad1357)

13. Compare the current work with previously published reports for AgNPs in separate table

14. Conclusions should be rewrite

15. Major English edition is required for whole manuscript

6. PLOS authors have the option to publish the peer review history of their article (what does this mean? ). If published, this will include your full peer review and any attached files.

**Do you want your identity to be public for this peer review?** For information about this choice, including consent withdrawal, please see our Privacy Policy .

Reviewer #1: No

Reviewer #2: No

---

## [Author Response · Author response to Decision Letter 1]

10 Feb 2025

Response to Reviewers comment

The authors thank all of the reviewers for their valuable insights and comments on the manuscript With title “Rhus vulgaris Meikle Fruit-Mediated Silver Nanoparticles: Synthesis, Characterization, and Potential for Sun Protection, Antioxidant and Antibacterial Applications ”. The reviewers’ comments have been taken into consideration and the manuscript has been revised properly accordingly. The revised manuscript shows the changes with red color text for your purview and kind consideration. The response to each comment has also been given below

Reviewers' comments

Reviewer's Responses to Questions

1. Is the manuscript technically sounds, and do the data support the conclusions?

Reviewer #1: Yes

Response for reviewer 1 comment: Thank you for your comments

Reviewer #2: Partly

Response for reviewer 2 comment: In our study, we have implemented rigorous experimental designs, including appropriate controls, adequate replication, and sufficient sample sizes to ensure the reliability and validity of our findings. Each experiment was carefully designed to minimize bias and variability, allowing us to draw robust conclusions based on the data collected. we made our conclusion is in line with the data

2. Has the statistical analysis been performed appropriately and rigorously?

Reviewer #1: Yes

Response for reviewer 1 comment: Thank you for your comments

Reviewer #2: Yes

Response for reviewer comment #1: Thank you for your comments

3. Have the authors made all data underlying the findings in their manuscript fully available?

Reviewer #1: Yes

Response for reviewer 1 comment: Thank you for your comments

Reviewer #2: No

Response for reviewer comment #2: Thank you for your comments regarding the PLOS Data policy. we have included all relevant data as part of the manuscript's supporting information. This includes not only summary statistics but also the individual data points that contribute to the means, medians, and variance measures presented in our findings.

4. Is the manuscript presented in an intelligible fashion and written in standard English?

Reviewer #1: Yes

Response for reviewer 1 comment: Thank you for your comments

Reviewer #2: No:

Response for reviewer comment #1: We have thoroughly reviewed the manuscript to identify and correct any typographical or grammatical errors. We have made revisions to ensure that the language is clear, correct, and unambiguous throughout the text

5. Review Comments to the Author

(a) Reviewer 1 comments

Dear Author, the manuscript entitled Rhus vulgaris Fruit-Mediated Silver Nanoparticles: Synthesis, Characterization, and Potential for Sun Protection, Antioxidant and Antibacterial Applications was written good but still needs to be revised with the following points.

Comment #1: The abstract can be modified and rewritten.

Response for comment #1: As per comments corrections are made on the manuscripts line number 19 to 39

Comment #2: The significance of R. vulgaris can be focused in the first part of the introduction.

Response for comment #2: Thank you for your suggestion regarding the placement of information about the significance of R. vulgaris in the introduction. As per comments corrections are made on the manuscripts line number 50 to 57

Comment #3: The exploration of AgNPs can be effectively written in the introduction with specific biological properties.

Response for comment #3: Thank you for your insightful comment regarding the exploration of silver nanoparticles (AgNPs) and their specific biological properties. As per comments corrections are made on the manuscripts line number 76 to 86.

Comment #4: Subheadings plant collection and preparation of plant extracts can be merged together and write in simplified manner.

Response for comment #4: As per comments corrections are made on line number 95 to 102. We marge three sub topics together and include under subtitle sample collection and preparation. we removed unnecessary additional information.

Comment #5: Remove the subheadings of characterization part and merge together.

Response for comment #5: As per comments corrections are made on line number 138 to 154. We marge all characterization techniques together and include under subtitle Characterization Techniques. We removed unnecessary additional information

Comment #6: Why has the author done only one antioxidant activity?

Response for comment #6: Thank you very much for your comments. We used only one antioxidant assay such as DPPH is due to its simplicity, reliability, and relevance to the study's goals, as well as practical considerations regarding resources and time.

Comment #7: The optimization part can be emphasized more with some more factors.

Response for comment #7: Thank you for your valuable feedback regarding the optimization section of my work. I appreciate your insights and would like to take this opportunity to elaborate on the optimization aspects and the factors that contribute to its synthesis silver nanoparticles. To synthesize silver nanoparticles, several critical parameters must be optimized, including pH, temperature, concentration, and reaction time. Our research work of existing research has highlighted the significant impact of these factors on the synthesis process. Therefore, we have incorporated these optimization parameters into our study to ensure the effective and efficient production of silver nanoparticles

Comment #8: The discussion related with AgNPs towards skin related research can be discussed more.

Response for comment #8: Thank you very much for the comments. As per comments we include the feedbacks in line number 460 to 472

Comment #9. Check for overall grammatical errors and sentence framing.

Response for comment #9: We have thoroughly reviewed the manuscript to identify and correct any typographical or grammatical errors. We have made revisions to ensure that the language is clear, correct, and unambiguous throughout the text

(b) Reviewer #2 comments

Authors presented “Rhus vulgaris Fruit-Mediated Silver Nanoparticles: Synthesis, Characterization, and Potential for Sun Protection, Antioxidant and Antibacterial Applications”, however, it requires more characterization part.

Authors should include and explain the below comments in the revised manuscript

Comment #1: Abstract should be rewrite, and should summarize the reason for the work, the most significant results, and the conclusions.

Response for comment #1: As per comments corrections are made on line number 19 to 39.

Comment #2: In introduction part, several categories of metallic NPs by green synthesis such as Au, Ag, Zno, Pd, TiO2, Cu, etc, have been reported with improved biomimetic attributes. Include these metal and metal oxide NPs with reference, Au (Phytosynthesis of gold nanoparticles using Caesalpinia pulcherrima (peacock flower) flower extract and evaluation of their antimicrobial activities) Ag (https://doi.org/10.1016/j.matpr.2020.10.048), ZnO and Cu (https://doi.org/10.1007/s43630-022-00224-0), Fe and Pd (DOI: 10.1039/C5CY00099H), etc.

Response for comment #2: Thank you for the commons. As per comments corrections are made on line number 77 to 87

Comment #3. Synthetic part: Why the authors used Rhus vulgaris Fruit extract only instead of other extract?

Response comment #3: Thank you very much for the comment. Since there hasn't been any earlier research on using Rhus vulgaris fruit extract to synthesize silver nanoparticles, we chose it. Rhus vulgaris is also known for its distinct phytochemical profile, traditional therapeutic use, and antioxidant qualities. Our selection of this specific extract is further supported by its accessibility and applicability to the goals of our investigation.

Comment #4: So many reports already published using the Rhus vulgaris Fruit extracts for the synthesis of AgNPs and biomedical application studies?

Response for comment #4: Thank you very much for your comment. There are several research works on Rhus vulgaris Fruit extracts and its biomedical application. However, as much as we know there is no research works on biomedical application using silver nanoparticle mediated by fruit extract of Rhus vulgaris for sun screening application, antioxidant and antibacterial activities

Comment #5. UV and IR spectral image should be clear, IR is not clear better to remove it

Response for comment #5: As per comments corrections are made. We changed the UV, and IR peaks to be clear figure 3 and 4 (line number 354 and 374)

Comment #6: Characterization part: all figures including IR, TEM, should be clear

Response for comment #6: As per comments corrections are made. We changed the IR, and TEM peaks to be clear.(line number 449)

Comment #7: XRD for AgNPs required with clear hkl values (111, 200, 220, 311, 222 for AgNPs) and explain the hkl values

Response for comment #7: As per comments corrections are made. We included hkl vale (111, 200, 220, 311, 222 for AgNPs) in XRD peaks in Figure 6 line 402 and in the text line number 389 to 392

Comment #8: Single particle or HRTEM should be included with EDAX

Response for comment #8: Thank you very much for nice comments. Because of the instruments available, for particle size determination, we utilized transmission electron microscopy (TEM), and to detect the presence of silver (Ag) metal, we employed UV-Vis spectroscopy due to the availability of instruments. In our future work we will include this nice comments

Comment #9: Author should explain how the average size of AgNPs is 10 nm to 20 nm?

Response for comment #9: Thank you very much for the comments. Using TEM image and imaje software the average size nanoparticle was calculated. This is indicated in line number 440 to 441

Comment #10: In addition, the lattice fringe spacings or D spacing hkl should be included

Response for comment #10: As per comments corrections are made on line number 396 to 397

Comment #11: For antioxidant activity: author performed DPPH, what about ABTS, and NOx radical scavenging analyses? Compare and refer (https://doi.org/10.2147/IJN.S210517)

Response for comment #11: Thank you very much for your comments. We used only one antioxidant assay such as DPPH is due to its simplicity, reliability, and relevance to the study's goals, as well as practical considerations regarding resources and time. As per comments corrections are made on line number 546 to 552

Comment #12. For antibacterial activity refer (DOI 10.1088/2053-1591/ad1357)

Response for comment #12: Thank you for your comments. As per comments corrections are made in line number 635 to 641

Comment #13. Compare the current work with previously published reports for AgNPs in separate table

Response for comment #13: As per corrections are made. We added new table (Table 3) that compare current work with other result in line number 404 to 433

Comment #14. Conclusions should be rewrite

Response for comment #14: As per comments corrections are made on line number 656-674

Comment #15. Major English edition is required for whole manuscript

Response for comment #15: We checked the English edition throughout the paper

---

## [Decision Letter · Decision Letter 1]

2 May 2025

Rhus vulgaris Meikle Fruit-Mediated Silver Nanoparticles: Synthesis, Characterization, and Potential for Sun Protection, Antioxidant and Antibacterial Applications

PONE-D-24-53986R1

Dear Dr. Limenew Abate Worku

We’re pleased to inform you that your manuscript has been judged scientifically suitable for publication and will be formally accepted for publication once it meets all outstanding technical requirements.

Kind regards,

Chellasamy Panneerselvam

Academic Editor

PLOS ONE

**Additional Editor Comments (optional):**

**Comments from PLOS Editorial Office** : We note that one or more reviewers has recommended that you cite specific previously published works. As always, we recommend that you please review and evaluate the requested works to determine whether they are relevant and should be cited. It is not a requirement to cite these works. We appreciate your attention to this request.

**Reviewers' comments:**

Reviewer's Responses to Questions

**Comments to the Author**

1. If the authors have adequately addressed your comments raised in a previous round of review and you feel that this manuscript is now acceptable for publication, you may indicate that here to bypass the “Comments to the Author” section, enter your conflict of interest statement in the “Confidential to Editor” section, and submit your "Accept" recommendation.

Reviewer #1: All comments have been addressed

Reviewer #2: All comments have been addressed

2. Is the manuscript technically sound, and do the data support the conclusions?

Reviewer #1: Yes

Reviewer #2: Yes

3. Has the statistical analysis been performed appropriately and rigorously? 

Reviewer #1: Yes

Reviewer #2: Yes

4. Have the authors made all data underlying the findings in their manuscript fully available?

Reviewer #1: Yes

Reviewer #2: Yes

5. Is the manuscript presented in an intelligible fashion and written in standard English?

Reviewer #1: Yes

Reviewer #2: Yes

6. Review Comments to the Author

Reviewer #1: The manuscript is written with adequate data related to the analysis carried out and is publishable.

Reviewer #2: The authors have adequately addressed all comments raised in a previous round of review and I feel that this manuscript is now acceptable for publication.

7. PLOS authors have the option to publish the peer review history of their article (what does this mean? ). If published, this will include your full peer review and any attached files.

**Do you want your identity to be public for this peer review?** For information about this choice, including consent withdrawal, please see our Privacy Policy .

Reviewer #1: No

Reviewer #2: No

---

## [Editor Report · Acceptance letter]

PONE-D-24-53986R1

PLOS ONE

Dear Dr. Abate,

I'm pleased to inform you that your manuscript has been deemed suitable for publication in PLOS ONE. Congratulations! Your manuscript is now being handed over to our production team.

Kind regards,

on behalf of

Dr. Chellasamy Panneerselvam

Academic Editor

PLOS ONE